# Structure, Diversity, and Environmental Determinants of High-Latitude Threatened Conifer Forests

Carlos Esse [1,*], Francisco Correa-Araneda [1], Cristian Acuña [1], Rodrigo Santander-Massa [1], Patricio De Los Ríos-Escalante [2,3], Pablo Saavedra [4], Ximena Jaque-Jaramillo [1], Roberto Moreno [1], Paola Massyel García-Meneses [5] and Daniel P. Soto [6]

1 Instituto Iberoamericano de Desarrollo Sostenible (IIDS), Unidad de Cambio Climático y Medio Ambiente (UCCMA), Universidad Autónoma de Chile, Avenida Alemania 01090, Temuco 4780000, Chile; francisco.correa@uautonoma.cl (F.C.-A.); cristian.acuna.loyola@gmail.com (C.A.); rodrigo.santander@uautonoma.cl (R.S.-M.); ximena.jaque.jaramillo@gmail.com (X.J.-J.); roberto.moreno@uautonoma.cl (R.M.)
2 Departamento de Ciencias Biológicas y Químicas, Facultad de Recursos Naturales, Universidad Católica de Temuco, Avenida Rudecindo Ortega N° 02950 Campus San Juan Pablo II, Temuco 4780000, Chile; prios@uct.cl
3 Núcleo de Estudios Ambientales, Universidad Católica de Temuco, Temuco 4780000, Chile
4 Facultad de Ciencias Forestales y Recursos Naturales, Escuela de Graduados, Universidad Austral de Chile, Valdivia 5090000, Chile; psaavedra@proyectos.uct.cl
5 Laboratorio Nacional de Ciencias de la Sostenibilidad (LANCIS), Instituto de Ecología, Universidad Nacional Autónoma de México, Ciudad de México 04510, Mexico; paola.garcia@ecologia.unam.mx
6 Departamento de Ciencias Naturales y Tecnología, Universidad de Aysén, Coyhaique 5950000, Chile; daniel.soto@uaysen.cl
* Correspondence: carlos.esse@uautonoma.cl

**Abstract:** *Pilgerodendron uviferum* (D. Don) Florin is an endemic, threatened conifer that grows in South America. In the sub-Antarctic territory, one of the most isolated places in the world, some forest patches remain untouched since the last glaciation. In this study, we analyze the tree structure and tree diversity and characterize the environmental conditions where *P. uviferum*-dominated stands develop within the Magellanic islands in Kawésqar National Park, Chile. An environmental matrix using the databases WorldClim and SoilGrids and local topography variables was used to identify the main environmental variables that explain the *P. uviferum*-dominated stands. PCA was used to reduce the environmental variables, and PERMANOVA and nMDS were used to evaluate differences among forest communities. The results show that two forest communities are present within the Magellanic islands. Both forest communities share the fact that they can persist over time due to the high water table that limits the competitive effect from other tree species less tolerant to high soil water table and organic matter. Our results contribute to knowledge of the species' environmental preference and design conservation programs.

**Keywords:** sub-Antarctic; Patagonia; *Pilgerodendron uviferum*; biodiversity indexes; Kawésqar National Park; WorldClim; SoilGrids

## 1. Introduction

The coastal temperate forests of southwestern South America are considered one of the 25 hotspots for biodiversity conservation [1]. This is due to their high level of endemism (ca. 50%) and the dramatic reduction of their original forest area (ca. 70%) due to human impacts [2,3]. This region is home to three endemic and monotypic Cupressaceae family members, and all have conservation issues [4,5]. One of them, the ciprés de las Guaitecas (*Pilgerodendron uviferum* (D. Don.) Florin), is the world's southernmost conifer and covers the greatest latitudinal distribution range (39°35′–54°20′ S; ca. 1600 km) [6] in the temperate region of South America. Specifically, this dioecious conifer dominates

Chile's sub-Antarctic forests and Magellanic moorlands and occupies poorly drained soils with high water tables [7–12]. *Pilgerodendron uviferum* has been intensely exploited since the beginning of the 17th century [13] due to the excellent quality of its wood [14]. Forest fires have significantly reduced the area where these forest communities develop [8,15]. To date, there is no accurate estimate of the surface burned, but some experts estimate that up to 90% of the area where this species is present has been impacted by fire, including remote areas in the southern Chilean archipelago. Due to these threats, *P. uviferum* has been classified as vulnerable by the IUCN (World Conservation Union) [5,16,17] and has been included in CITES Convention Appendix 1, which regulates international trade of threatened species.

Based on research conducted in southwestern Chile, specifically in the Magellanic islands of Puerto Eden (49°08′ S), [18] described *P. uviferum* stands as turbulent forests typical of the rainy temperate coastal zone and as one of the least studied forests in South America [19]. According to the literature [8], primary succession in the southern Magellanic region usually begins in flat sectors without drainage, where the initial species are *Sphagnum* spp. and grasses. These species generate peatlands, and better soil drainage allows *P. uviferum* seedlings to establish. These areas are called "bog forests" [12]. This species can develop in wet and poorly drained soils, which means that it presents oxide reduction processes on the mineral horizon, generating gleysols with a large amount of organic matter, mostly due to the presence of peat moss (*Sphagnum* spp.) [6,8,11,12,20].

These communities then transition from bog forests to open cypress forests with uneven-aged structure and abundant regeneration in the absence of major disturbances [21]. Late successional forests are dominated by *P. uviferum* and characterized by having open to very open canopy conditions, which often present around 10% cover [8,22,23]. In flat areas with poor soil drainage, forests dominated by *P. uviferum* form a hydrosere [23] due to the accumulation of organic matter, mainly from *Sphagnum* spp., which is accompanied by abundant regeneration with high overstory mortality [6,23]. The species has faced various threats due to human disturbances over a large part of its territory [23], including pre-Hispanic burns that reduced and degraded the species' habitats [24]. As a result, *P. uviferum* regenerates poorly and may be replaced by other species, such as *Drimys winteri* (J.R. and G. Forst.), *Tepualia stipularis* (Hook. and Arn.) Griseb., and *Saxegothaea conspicua* (Lindl). In the case of low-intensity fires, *P. uviferum* regeneration is abundant and can be characterized as an open peat forest dominated by *P. uviferum* and in the lower canopies dominated by *T. stipularis* and *Nothofagus betuloides* (Mirb.) Oerst., covering less than 30% of the surface area [8,11,12].

Research conducted in remote Magellanic islands, especially in the northern part of Kawésqar National Park (KNP), [21] has proposed a classification of vegetation using the so-called "vegetation floors". This corresponds to a standardized cartographic classification of Chilean vegetation based on a 1 km² grid. Forests in which *P. uviferum* is the main tree component are classified as coastal antiboreal peatlands [21]. *P. uviferum* has become a dominant tree in the forests, and they share the overstory with *T. stipularis* and *N. betuloides* in locations covered by peat and bog that are less waterlogged. However, very little is known about these forests due to intense human disturbances along their distribution range [11,12,21]. Some studies on this species have revealed important changes and impacts due to stressors, such as wind, fire, climate change, and glaciation [19,25,26]. Our hypothesis is that forests on Magellanic islands present different structures, community composition, and diversity that are influenced by climate (temperature and precipitation) and edaphic variables (water and carbon), where the main driver among the stands is the high soil water table. This study thus contributes to an ecological understanding that can inform the design of conservation programs in the southern cone of the American continent.

## 2. Materials and Methods

### 2.1. Study Area

The study area is located within Kawésqar National Park, the insular part of the provinces of Última Esperanza and Magallanes in southwestern Chile (Figure 1). The climate is considered a hyperoceanic antiboreal bioclimate with humid to hyperhumid soils [21]. The local climate is isothermal tundra and presents a mean annual temperature of 6.5 °C [20]. Annual rainfall can reach up to 6000 mm, making it one of the rainiest areas in the world [27,28]. The presence of *Pilgerodendron uviferum* occurs from Valdivia Province (39° S) to Magallanes Province (54° S) [29,30].

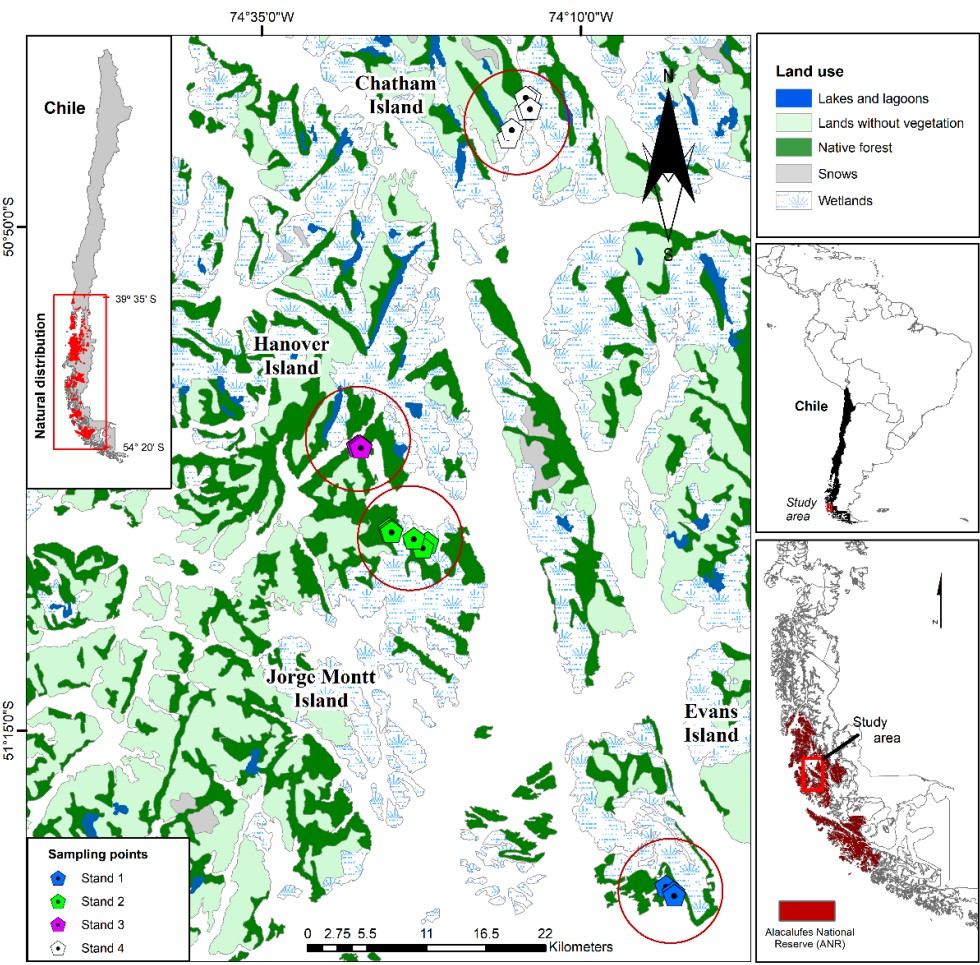

**Figure 1.** General location of the study area and sample units established in Kawésqar National Park (KNP), Chile.

These soils present great fragility and susceptibility to degradation [31]. They also have a low concentration of $O_2$ and high Al and Fe. Their pH ranges from 3.5 to 5, they have a high water table, and they are generally classified as gleysols and histosols [8,28].

### 2.2. Sampling and Data Collection

Four stands without apparent human disturbances were selected in the insular area of KNP (e.g., without the presence of domestic animals (dung), tree logging (stumps), and hiking trails). Simple random sampling was performed in each stand. We sampled a total of 18 sample units (plots) within the stands. These sampling units were circular plots of 500 $m^2$, where the diameter at breast height (DBH), total height (m), and associated vertical position for dominant, codominant, intermediate, submerged, and suppressed trees were

recorded [6]. In each plot, we considered as trees those individuals with a DBH equal to or greater than 5 cm.

The quantitative overstory structure of the stands was described in terms of density (D), mean square diameter (MSD), average height ($H_{avg}$), and total basal area (BA). In addition, a circular and concentric subsampling unit of 2 m radius was established in each sample unit ($n = 18$) to quantify tree regeneration. The nomenclature of the tree species was based mainly on literature descriptions [32], and its phytogeographic origin followed [33]. To determine the age of the stands, the annual growths of three dominant individuals in terms of tree height were sampled per plot. Two increment cores were obtained at 0.3 m above ground using Pressler's borer. The relationship between age and DBH was assessed through simple linear regression [34]. Coefficient of determination ($r^2$), standard deviation, and structure of residuals were visually evaluated. The cores were processed following standard tree ring procedures [35] and measured using WinDendro$^{TM}$ software (Regents Instruments Inc., Québec, QC, Canada).

## 2.3. Diversity

A diversity analysis was conducted in each plot based on the richness index (S), the number of tree species in each stand [36]; total abundance (N), density of trees per plot; and Shannon–Wiener diversity index (H′), which indicates the uniformity of the values across all stand species [37]. The indices were obtained using PAST 3 [38] and the Biodiversity package in R software [39].

## 2.4. Environmental Matrix Based on Biophysical Variables

To identify the environmental variables that explain the presence of *P. uviferum* in the stands, an environmental matrix was built using climate, soil, and topography variables. The variables selected are listed in Table 1. Climate information was obtained from the WorldClim database [40]. The WorldClim data consist of raster layers generated using the interpolation of average monthly climate based on data from weather stations. A 30-arc-second resolution grid is provided, which includes precipitation and temperature data for the entire world. On the other hand, soil variables were obtained from the SoilGrids database [41], which display basic soil properties based on the FAO world reference, and the USDA soil taxonomy suborders in raster format for the whole world. The geographic coordinates at the center of each sampling point were recorded using a GPS instrument (Garmin map 64sx). Information from WorldClim and SoilGrids was extracted using the vector coverage of sampling points in a spatial analysis using the GIS ArcMap v10.8.1 software. Finally, the topographic variables were obtained from the "catastro de los recursos vegetacionales nativos de Chile" database [42].

## 2.5. Statistical Analyses

The nonparametric Kruskal–Wallis (*H*-test) test was used to identify significant differences between the stand parameters and the diversity indices at a level of 95%. We previously checked the parametric assumptions (i.e., normality and variance homoscedasticity), which were not met for all variables used. A Wilcox post hoc test was run when significant differences were obtained from the *H*-test ($p < 0.05$) with the Holm method to correct the level of significance [43,44]. This analysis was carried out with Kruskal function in the R package agricolae [45].

Principal components analysis (PCA) with a correlation matrix was performed to identify the environmental variables that best explain the variability of the data set along the multivariate space (i.e., axes) [34]. The R packages factoextra [46] and FactoMineR [47] were used for this purpose. Finally, Pearson correlation analysis was run between the main PCA axes with the reduced environmental variables to interpret the main environmental gradients driving the *P. uviferum*-dominated forest communities.

**Table 1.** Factors and environmental variables analyzed (soil variables estimated at a depth of 0.3 m).

| Factor | Variable | Unit | Description | Reference |
|---|---|---|---|---|
| 1. Climate | AMT | °C | Annual mean temperature | Hijmans et al. [40] |
| | MDR | °C | Mean diurnal range (mean of monthly (max temp−min temp)) | |
| | ISO | Index | Isothermality (BIO2/BIO7) (×100) | |
| | TSS | Index | Temperature seasonality (standard deviation × 100) | |
| | MTW | °C | Max temperature of warmest month | |
| | MTC | °C | Min temperature of coldest month | |
| | TAR | °C | Temperature annual range (BIO5–BIO6) | |
| | MTQ | °C | Mean temperature of wettest quarter | |
| | MTD | °C | Mean temperature of driest quarter | |
| | MWQ | °C | Mean temperature of warmest quarter | |
| | MCQ | °C | Mean temperature of coldest quarter | |
| | APP | mm | Annual precipitation | |
| | PWM | mm | Precipitation of wettest month | |
| | PDM | mm | Precipitation of driest month | |
| | PPS | % | Precipitation seasonality (coefficient of variation) | |
| | PWQ | mm | Precipitation of wettest quarter | |
| | PDQ | mm | Precipitation of driest quarter | |
| | PPW | mm | Precipitation of warmest quarter | |
| | PCQ | mm | Precipitation of coldest quarter | |
| 2. Edaphic | PWP | % | Available soil water capacity (volumetric fraction) until wilting point | SoilGrids Hengl et al. [41] |
| | Silt | % | Silt content (2–50 micrometer) mass fraction | |
| | Sand | % | Sand content (50–2000 micrometer) mass fraction | |
| | OC | kg m$^{-3}$ | Soil organic carbon density | |
| | C | permille | Soil organic carbon content | |
| | pH | scale | Soil pH × 10 in $H_2O$ | |
| | CEC | Cmol kg$^{-1}$ | Cation exchange capacity of soil | |
| | Clay | % | Clay content | |
| | BD | kg m$^{-3}$ | Bulk density | |
| | SD | cm | Absolute depth to bedrock | |
| 3. Topographic | Aspect | range | Aspect (rank) | CONAF et al. [42] |
| | Slope | % | Slope (degree in %) | |
| | Elevation | m asl | m asl (meter above sea level) | |

To evaluate potential differences among communities (stands), we conducted the permutational multivariate analysis of variance (PERMANOVA) with a post hoc test using the Bray–Curtis distance [48,49]. This analysis was performed with 1000 permutations with significant *p*-values > 95%. To visualize stands in the multivariate space, nonmetric multidimensional scaling (nMDS) analysis was performed using the original abundance matrix. Analysis was carried out using the adonis function in the R package vegan [50].

## 3. Results

### 3.1. Quantitative Structure of the Stands

The dasometric parameters showed differences between stands. Stand 1 had the lowest average dominant height, followed by Stands 3 and 4. Stand 2 had the bigger trees in terms of quadratic mean diameter ($p < 0.05$, Table 2). The oldest stands were Stands 3 (245 years) and 2 (242 years), and the youngest were Stands 4 (201 years) and 1 (214 years) ($p < 0.05$, Table 2).

The diameter distribution showed that Stand 2 had the most developed forest structure, unlike the stands that had greatest density in the first diameter classes. This suggests that there is a trend towards reverse-J diameter distribution based on evidence that a larger number of individuals are in the smaller size classes (Figure 2).

**Table 2.** General background of the stands measured. $H_{avg}$ = average height; MSD = mean square diameter; D = density; BA = basal area.

| Parameters | | Stands | | | | p-Value |
|---|---|---|---|---|---|---|
| | | S1 | S2 | S3 | S4 | |
| Descriptive | Aspect | SW | FLAT | SW | FLAT | - |
| | Area (ha) | 41.83 | 175.71 | 22.31 | 112.23 | - |
| | Age (years) | 214 (1.0) [c] | 242 (0.5) [b] | 245 (1.0) [a] | 201 (1.0) [d] | 0.001 |
| Dasometric | $H_{avg}$ (m) | 5.80 (0.2) [d] | 11.00 (1.1) [a] | 6.30 (0.2) [c] | 7.20 (0.2) [b] | 0.001 |
| | MSD (cm) | 18.57 (4.3) [ab] | 25.13 (5.0) [a] | 17.40 (4.0) [b] | 18.20 (2.4) [b] | 0.044 |
| | D (trees ha$^{-1}$) | 806 (208.2) | 856 (290.0) | 1087 (477.0) | 1187 (290.0) | 0.237 |
| | BA (m$^2$ ha$^{-1}$) | 21.27 (6.8) | 40.43 (14.7) | 23.57 (7.1) | 30.10 (5.3) | 0.081 |

Note: Significant differences (Kruskal–Wallis test, $p < 0.05$) are indicated by different letters. The values are means and standard deviations (in brackets).

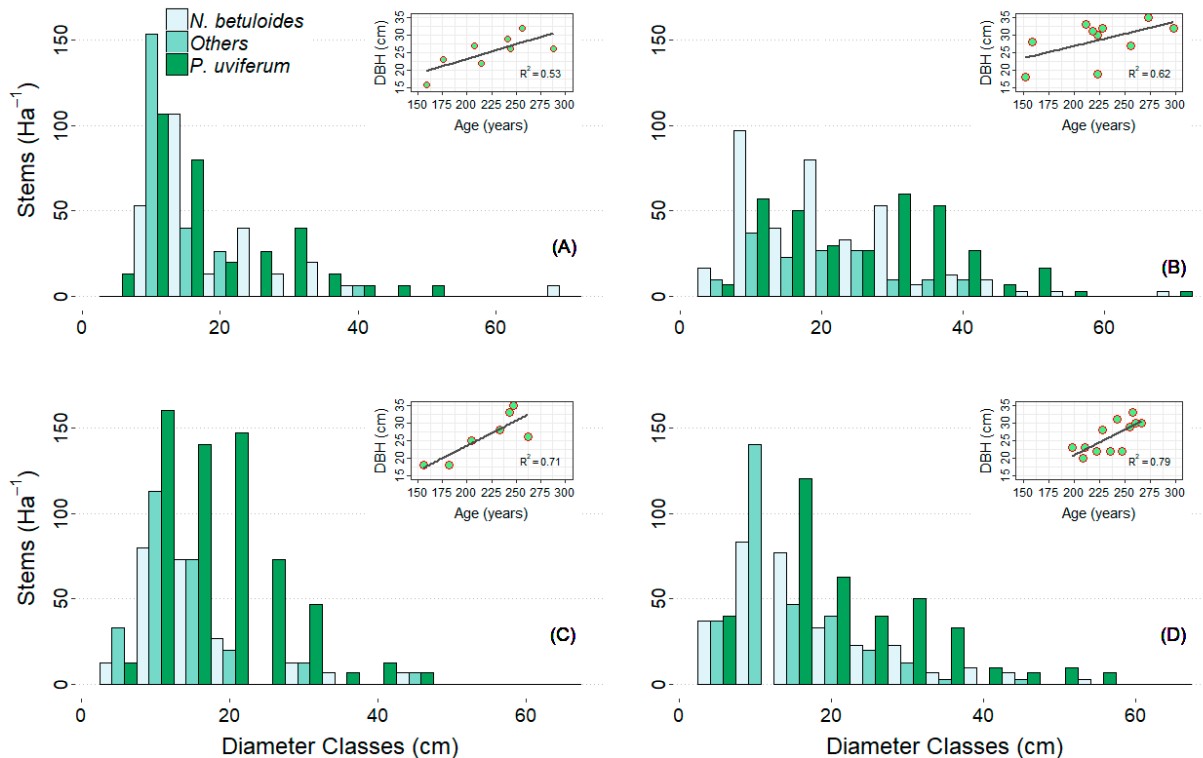

**Figure 2.** Diameter distribution and age diameter relationship per stand (inside panels). In (**A**) Stand 1, (**B**) Stand 2, (**C**) Stand 3, and (**D**) Stand 4.

### 3.2. Analysis of the Environmental Matrix Based on Biophysical Variables

The PCA suggests that the first three axes accounted for 78.44% of the explained variance, reducing the 33 variables to the three main dimensions with 16 variables. Specifically, the first axis accounted for 40.34% of the explained variance and was represented by the climate factor (APP, PWP, PDM, PWQ, PDQ, PPW, and PCQ), in this case related to precipitation variables. The second axis accounted for 22.33% of the explained variance and was represented by the climate (MTC, MTD, and MCQ), topographic (elevation), and soil factors (C and OC), and in this case the greatest contribution was made by elevation ($r = 0.348$). The third axis accounted for 15.77% of the explained variance represented by the climate factor (MTW, MWQ, and MTQ), in this case related to the mean annual temperature of the warmest quarter ($r = 0.354$) (Table 3). Figure 3 depicts the biplot obtained by PCA, and Table 3 shows the correlation ($r$) between selected environmental variables with the first three dimensions.

**Table 3.** Pearson correlation analysis (*r*) of environmental variables with the first three principal components (selected variables appear in italics and bold).

| Variable | PC1 | PC2 | PC3 |
|---|---|---|---|
| Aspect | −0.034 | 0.180 | 0.006 |
| Slope | −0.028 | 0.135 | 0.208 |
| *Elevation* | −0.002 | **0.348** | −0.117 |
| AMT | −0.023 | −0.257 | 0.296 |
| MDR | 0.205 | 0.169 | 0.199 |
| ISO | 0.160 | 0.120 | 0.013 |
| TSS | 0.201 | 0.132 | 0.234 |
| *MTW* | 0.141 | −0.053 | **0.354** |
| *MTC* | −0.152 | **−0.288** | 0.103 |
| TAR | 0.208 | 0.157 | 0.193 |
| *MTQ* | −0.160 | −0.005 | **0.317** |
| *MTD* | −0.193 | **−0.256** | 0.010 |
| *MWQ* | 0.062 | −0.181 | **0.356** |
| *MCQ* | −0.118 | **−0.297** | 0.160 |
| *APP* | **−0.273** | 0.008 | −0.008 |
| *PWM* | **−0.272** | −0.001 | −0.038 |
| *PDM* | **−0.272** | −0.017 | −0.029 |
| PPS | −0.229 | 0.130 | 0.120 |
| *PWQ* | **−0.273** | 0.003 | −0.014 |
| *PDQ* | **−0.273** | 0.008 | −0.003 |
| *PPW* | **−0.272** | 0.010 | 0.000 |
| *PCQ* | **−0.273** | 0.004 | −0.010 |
| Tex | −0.100 | 0.216 | 0.114 |
| PWP | 0.150 | −0.229 | −0.207 |
| Silt | 0.185 | −0.141 | −0.130 |
| Sand | −0.116 | 0.204 | 0.168 |
| *OC* | 0.094 | **−0.259** | −0.124 |
| *C* | 0.116 | **−0.278** | −0.058 |
| pH | −0.199 | 0.141 | −0.043 |
| CEC | 0.057 | 0.033 | −0.296 |
| Clay | −0.013 | −0.196 | −0.125 |
| BD | −0.063 | 0.068 | −0.244 |
| SD | −0.036 | −0.138 | 0.182 |

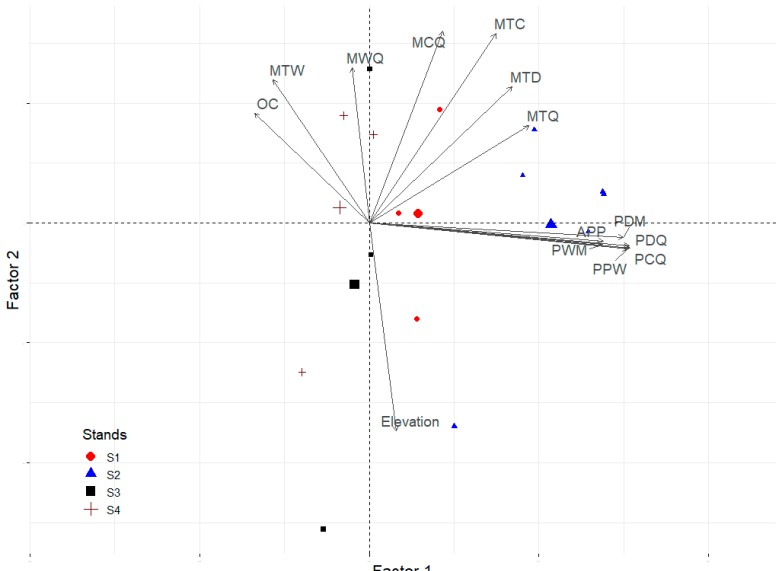

**Figure 3.** PCA biplot based on the main environmental variables extracted from the WorldClim and SoilGrids databases and topographic variables per stand.

### 3.3. Composition, Diversity, and Community Structure

*Pilgerodendron uviferum*, *Tepualia Stipularis* (Hook et Arn.) Griseb, *Drimys Winteri* J.R. et G. Forster., *Maytenus Magellanica* (Lam.) Hook. F., *Nothofagus Betuloides* (Mirb) Oerst., *Pseudopanax laetevirens* K. Koch., *Nothofagus antarctica* (G. Forst.) Oerst., and *Podocarpus nubigenus* Lindl. were found in this study. Regarding tree species that represent the major overstory composition (n) for each stand, nine were identified in Stand 4, six in Stand 2, six in Stand 1, and five in Stand 3. *P. uviferum* was the most abundant tree species in all stands, followed by *N. betuloides* and *T. stipularis* (Table 4).

**Table 4.** Abundance of species in each stand in Kawésqar National Park. The values are the means and standard error (in brackets) per stand.

| Species | Abundance by Stand | | | |
| --- | --- | --- | --- | --- |
| | Stand 1 | Stand 2 | Stand 3 | Stand 4 |
| *P. uviferum* | 16.33 (6.89) | 26.68 (10.74) | 35.70 (16.37) | 32.19 (7.33) |
| *T. stipularis* | 39.53 (22.54) | 1.17 (0.98) | 2.67 (2.19) | 17.41 (11.74) |
| *D. winteri* | 2.33 (1.45) | 4.17 (2.43) | 10.00 (4.93) | 9.83 (2.30) |
| *L. ferruginea* | - | - | - | 2.50 (1.23) |
| *M. magellanica* | 0.33 (0.33) | 1.33 (1.33) | - | 0.67 (0.33) |
| *N. betuloides* | 13.00 (3.51) | 25.38 (3.74) | 13.35 (1.86) | 17.02 (1.06) |
| *P. laetevirens* | 2.35 (1.46) | 4.70 (3.73) | 6.38 (5.88) | 3.18 (1.46) |
| *N. antarctica* | - | - | - | 2.00 (0.86) |
| *P. nubigena* | - | - | - | 2.33 (1.12) |

Significant differences among the stands were observed in terms of species richness (S′) and Shannon index ($p < 0.05$, Table 5). For example, the Shannon index values showed low diversity. Stand 4 was the most diverse, with a value of 1.45, and Stand 2 was the least diverse, with a value of 0.89. There were significant differences between the stands in terms of species richness. Stand 4 was the most diverse, followed by Stands 1, 2, and 3, which were similar. No significant differences were found among the stands in terms of abundance.

**Table 5.** Diversity indices for species in different stands in Kawésqar National Park.

| Index | Diversity Index by Stand | | | | *p*-Values |
| --- | --- | --- | --- | --- | --- |
| | Stand 1 | Stand 2 | Stand 3 | Stand 4 | |
| Richness (S′) | 4.67 (0.33) [b] | 3.83 (0.40) [b] | 4.33 (0.33) [b] | 6.33 (0.21) [a] | 0.005 |
| Abundance (N) | 74.00 (32.13) | 63.33 (10.81) | 68.00 (19.00) | 87.17 (17.47) | 0.870 |
| Shannon index (H′ loge) | 1.21 (0.09) [ab] | 0.89 (0.14) [b] | 1.08 (0.22) [ab] | 1.45 (0.04) [a] | 0.028 |

Significant differences (Kruskal–Wallis test, $p < 0.05$) are indicated by different letters. The values are means and standard deviations (in brackets).

Finally, PERMANOVA shows a solution where the four stands under study form two main forest communities ($R = 0.53$, $p < 0.05$). *Post hoc* test showed significant differences ($p < 0.05$) between Stand 2 and Stands 1, 3, and 4, which were similar in terms of composition. The difference between two communities was confirmed by an nMDS biplot (Figure 4).

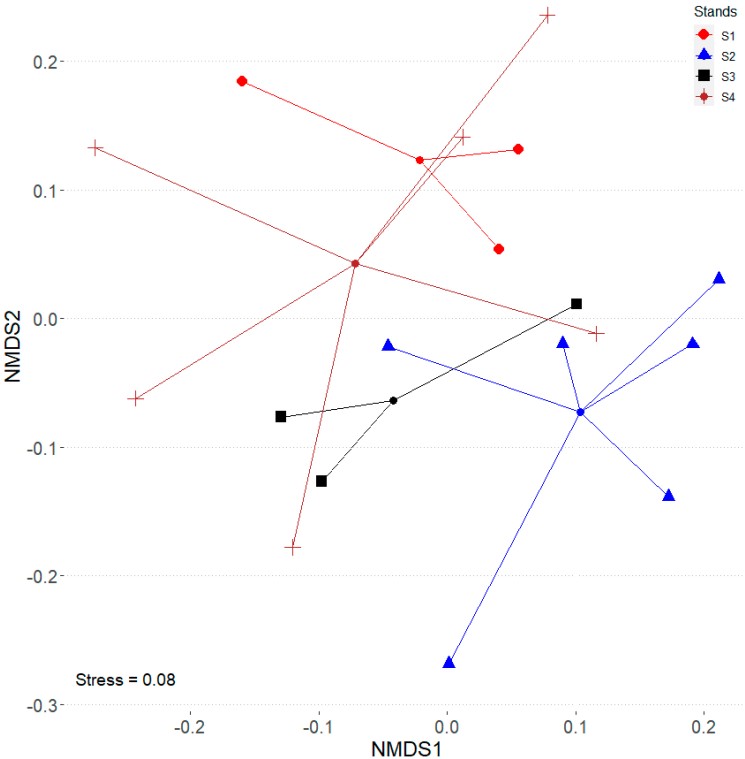

**Figure 4.** Nonmetric multidimensional scaling (nMDS) showing *Pilgerodendron uviferum* stands sampled in the Magellanic islands of Kawésqar National Park.

## 4. Discussion

This study contributes to our understanding of the structure, diversity, and dynamics of pristine forests dominated by *P. uviferum* and the environmental variables that influence its presence in the southwestern Patagonia archipelago. The results may help to expand and refine current knowledge about the ecology and dynamics of forest communities dominated by *P. uviferum*, since most studies have been conducted in the northern limit of its distribution range [11,30,31] and in some locations of southern Chiloé [8,12,15,51], where the soils and climate characteristics and even anthropogenic threats may cloud the interpretation of the ecology of these threatened conifer forests.

The forests dominated by *P. uviferum* have a wide latitudinal distribution, which could indicate that the species is capable of adapting to different environmental conditions if the water table remains high [11,12]. These forests are most abundant in the southernmost parts of Chile, especially in southern Chiloé and the Magellanic islands, where the mean annual precipitation is above 2000 mm [7,11,52]. In that way, climate seems to be the major factor driving the occurrence and abundance of these forests, where high rainfall and low temperature regimes are preferred by a *P. uviferum*-dominated forest community. In our study, we highlight that precipitation is the main variable driving the presence of *P. uviferum*-dominated communities, followed by soil variables, such as carbon and organic matter, but more importantly, the elevation above sea level. Finally, variables from the third group that explain the presence of *P. uviferum* are those related to temperature during the warmest months. These interpretations are supported by earlier studies [7,29].

Specifically, the climate factors influence the presence and abundance of *P. uviferum* communities on Magellanic islands. The high mean annual precipitation (up to 6000 mm) and low mean annual temperature are the main variables that explain the communities' distribution. We also found that high levels of soil carbon and/or organic material explain the presence of these communities on smaller scales. This is consistent with the studies conducted by [12,53] on Chiloé and the research performed at the northern edge of the coastal range of the Valdivian ecoregion [12]. However, the common variable among

all of the sites is the high soil water table, which is considered key for the recovery of *P. uviferum*-dominated communities after disturbance [8,12,24,53]. Therefore, soil water and organic matter amounts in conjunction with high amount of precipitation and low temperatures during the warmest months have an important influence on the structure and composition of these forest communities.

Stand 4 had a low water table that may have been improved through natural succession. One factor of this argument is the regeneration of conspecific tree species that are less tolerant of high water tables, such as *Nothofagus betuloides* and *Tepualia stipularis*, which was higher in that stand than in the rest [6]. The forest structure and composition of Stand 4 are comparable to the forest described in southern Chiloé Island by [8,54], who named it the "upland forest" due to the more favorable drainage conditions, which allow for irregular structures and mixed overstory composition. In waterlogged soils, *P. uviferum*-dominated communities tend to generate sparse and open forests, which coincide with the "open cypress forests" proposed by [8]. This forest type coincides with Stand 2 of our study, which presents greater tree species diversity, larger basal area, and the tallest and oldest trees included in our study (Table 2). This argument is consistent with the positive relationship between biodiversity and productivity on a global scale [55]. Moreover, its diameter distribution shows advanced stand development compared with the others, consistent with the maximum age registered for this stand (i.e., 374-year-old trees). Age–diameter relationship was positive and significant, consistent with previous studies conducted for *P. uviferum* in northern stands [12,54]. Stand 1 showed a similar trend to Stand 3 regarding age (263 and 288 years), but with shorter *P. uviferum* and a smaller basal area. The structure and composition of this stand were different from those of the rest, and they were similar to those described by [12,54]. This is reflected in the clear separation between Stands 1, 3, and 4 and Stand 2 (i.e., PERMANOVA and nMDS analysis).

Ecologically, stands without disturbances and in flat areas have extremely high soil water table and present lower soil temperature and low nutrient availability [12]. In these site conditions, *P. uviferum* may undergo root modification [12,56]. This may be attributed to species adaptation and avoidance of death by maintaining low growth rates [53]. This morphological modification can be seen as the species' approach for adapting to extreme soil conditions in order to live for extended periods of time (e.g., 566 to 886 years in age) [12]. Under these swampy conditions, *P. uviferum* presents lower heights and diameters compared with forests with lower soil water tables or those growing in upland conditions [12]. The upland conditions described by [12] are similar to those reported for Stands 2 and 4 in our study.

In a broader perspective, according to the classification proposed by [22], Stands 1, 2, and 3 are located on the P91 floor, which is characterized by lowest species diversity. This is consistent with our results. Stand 4 is located on vegetation floor P94, where *P. uviferum* is the main canopy species along with the shrub species *S. magellanicum*. In this sense, Stand 1 presented low density of *P. uviferum*, which may be associated with harsh macroclimate conditions. If these are characterized by low temperatures and high precipitation in conjunction with a high water table, conditions may produce mortality and open forests with high abundance of *Sphagnum* spp. This is consistent with the conditions reported by [8] and [12] in southern Chiloé.

*Implication for the Conservation of High-Latitude Conifer Forests*

Monitoring remote forests poses economic, technical, and scientific challenges [57], but these efforts reveal how ecological factors and pressures could impact the structure and diversity of *P. uviferum* stands on Magellanic islands. Many conservation programs are incorporating remote sensing and other GIS tools to monitor hard-to-reach ecosystems, and these tools can be useful for conducting research in remote areas. These tools have been used for UAV biodiversity monitoring of Mount Venere, Italy [58]. The methodology utilized by [59] was used in a temporal study of forest degradation and deforestation in Brazil [60]. In addition, and considering pressures from other global phenomena, such

as changes in precipitation and precipitation regimes and the intensification of extreme climatic events (e.g., droughts and heatwaves during summer and cold events and intense rainfall in winter), forests such as the ones studied here could serve as a proxy for environmental changes and as a reference to compare with other forests that have been degraded.

## 5. Conclusions

This study documented the structure, composition, and diversity of *Pilgerodendron uviferum*-dominated stands on Magellanic islands in Kawésqar National Park, located in in the southwestern part of South America. We identified two major community groups that differ in tree structure, composition, and diversity. They are influenced by contrasting precipitation, temperature, soil carbon, and elevation and share their dependence on high water tables in order to persist in these forest communities. The information gathered in this study may help to improve conservation and restoration programs and to refine ecological knowledge of this understudied forest type.

**Author Contributions:** C.E.: conceptualization, formal analysis, funding acquisition, methodology, visualization, and writing—review and editing. R.S.-M., P.M.G.-M., and D.P.S.: formal analysis, software, validation, and writing—original draft. R.M. and P.D.L.R.-E.: data curation and validation. P.S. and X.J.-J.: formal analysis and visualization. C.A.: data curation, funding acquisition, and investigation. F.C.-A.: funding acquisition, formal analysis, and visualization. All authors have read and agreed to the published version of the manuscript.

**Funding:** This research was funded by the National Forestry Corporation (CONAF) of Chile and FONDECYT Projects, grant numbers 11160650 and 11170390 of ANID Chile, and Project MECESUP UCT 0804.

**Data Availability Statement:** The data presented in this study are available on request from the corresponding author. The data are not publicly available due to privacy.

**Acknowledgments:** The authors would like to thank the CONAF (Chilean Forest Service) of Chile, especially the Última Esperanza Province office. They are grateful for the assistance of Cristian Ruiz Huichapani, Patricio Salinas Dillems, Giovanny Serey Caba, Jovito González Chambla, Captain German Coronado Vázquez, and the crew of the *Yepayek*. C.E. acknowledges the support received through FONDECYT No. 11160650. F.C.-A. acknowledges the support received through FONDECYT No. 11170390. R.S.-M. acknowledges the funding received for the fourth author's graduate studies from Human Capital Training Fund Project No. 21150903. The authors express their gratitude to M.I. and S.M.A. for their valuable suggestions to improve the manuscript.

**Conflicts of Interest:** The authors declare no conflict of interest.

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
