# Peer review of "Structure, Diversity, and Environmental Determinants of High-Latitude Threatened Conifer Forests"

_forests, doi:10.3390/f12060775_

Round 1
Reviewer 1 Report
This study explores the environmental conditions and ecological features of sub-Antartic forests containing a focal species: P. uviferum. Overall this study is a useful exploration of an under-studied region, with useful messages for environmental management that is likely to be well-regarded by interested parties.
I have some concerns about the manuscript in its current form. First, the authors need to take care in not misrepresenting their findings, stating that they analyze the “environmental preferences”. Their sampling only includes stands containing P. uviferum, not stands where it is absent, which limits their ability to inform environmental preferences or tolerances in the study region.
One of the major things missing from this study was an examination of the interplay between the PCA of climate and soil products and the field-collected forest composition data. We see three sites separated in PC1 by their high silt, organic carbon and soil water capacity. In contrast, the authors divide their four stands into two groups of two “forest types”. I could only find a specific discussion of stand 4, which highlighted its low water table, which doesn’t seem to match those three sites in the PCA (which appear to be water-logged, silt and carbon-rich soils. I would like to see more of a discussion around which stand this was, and how such different environments can provide such similar forest composition (as per the nMDS).
My final concern is the lack of detail in the methods, especially in the statistical analyses. There are sections where it would be impossible to recreate the author’s results without trial-and-error, as they do not specify how they treated their data prior to analysis. They also make several choices without justification, or with invalid justification (such as removing environmental variables prior to PCA). I’ve highlighted many of these in my line comments below.
Line comments:
Abstract: Much of the abstract is dedicated to listing the statistical analyses (lines 24-26).
18: “that grow” should be “that grows”.
21: They’re not really environmental preferences because you don’t sample places where P. uviferum isn’t, so you can’t capture the boundaries of environmental tolerances. Also environmental and biotic filtering are conflated, as you’re observing the outcome of these two factors in your surveys.
36-52: To be honest, I think the authors could remove this entire paragraph. It contributes nothing of value to the manuscript and the start of the next paragraph on line 53 is a perfect start to the authors’ actual study.
77: P. uviferum is not italicised.
91: ‘presents’ doesn’t fit the context. Perhaps “...P. uviferum regenerates poorly and may…” as an alternative.
100: 2 in km2 is not superscripted.
112: Being the first is not a valid justification for study. Remove this sentence.
124: 2 in O2 superscripted.
128: How was the ‘without apparent human disturbance’ decision made?
129-131: There seems to be a conflict between “simple random sampling” and “distributed proportionally”. Surely simple random sampling involves a process like random number generation to select sites, which may not be proportional. Some clarity here would be welcome.
129-131: Why were there six sampling sites in stands 2 and 4, but only 3 in stands 1 and 3? This presents a problem for the PCA, as the environmental space is not equally sampled, as some spatial clusters (stands) have twice as much sampling as others, which may have biased the PC loadings.
131-133: All trees? Including seedlings? Was there a minimum height or DBH below which trees weren’t surveyed?
135-137: These data are not included in the results or discussion. Either include them if they are relevant, or remove this sentence.
134: Unnecessary space between parentheses in (D )
Figure 1: This is a useful figure to understand the study site context. One thing I would like to see added is the distribution of P. uviferum, perhaps in the whole of South America panel on the side. The authors discuss the broad latitudinal distribution, it would be interesting to compare the extent of this to the study site.
155-165: It is unclear how the authors extracted data from Worldclim and other datasets. Did they use the coordinates of each sample site? It appears this way from the points in Figure 3 but the authors need to make this clearer. Also, why did the authors obtain longitude and latitude coordinates from a past study (lines 164-165)? Were these sites the same as were used in the past?
Table 1: The authors should consider not reporting BioClim variables using their number, as these numbers are meaningless, especially when reported in text as a list or in figures, both of which the authors do further in the manuscript.
171: What is the justification for using a non-parametric test here? Were there issues with parametric models? Kruskal-Wallis uses rank order rather than true values, and thus identify between group differences that are much smaller when considering the actual values. If there was no strong rationale for this test, I would suggest the authors re-analyze using mixed-effects linear regressions, as they will need to address the spatial clustering in their data collection (sites within stands) using random intercepts.
176: One of the major use cases for PCA is to remove collinearity, thus this is not a justifiable reason for removing environmental variables. Each principal component is an uncorrelated combination of input variables, so it doesn’t matter if they are uncorrelated or perfectly collinear. Authors should provide a better rationale for variable exclusion. Also, listing “bio” variables by number should be avoided.
180-183: Was the PCA run on the covariance matrix or the correlation matrix? I assume correlation, as variances of the input variables are unlikely to be equivalent, but the authors don’t state this.
183-185: Can the authors justify this choice? What does a Pearsons’ correlation show that the PCA eigenvector loadings don’t? If they choose to justify this choice, I’d also like some more detail on how this was performed.
190: There is no reference to the “unweighted average ligation technique” in Ref 64. Do you mean you didn’t weight the species occurrences? Did the authors correct for abundance by converting to relative abundance proportions? Did they square root abundances? Ref 64 discusses all of these options, and the authors don’t specify what they did.
Figure 2: Panel B should have the same y-axis limits as the other panels. Also “density” is not usually a unitless measurement (except in kernel density estimates (KDEs)). Are these units in stems ha-1 for instance? I also feel like the term “density curves” is inaccurate, as these appear to be histograms rather than modelled curves or KDEs.
232: “may be”? These are your results, you shouldn’t need to guess to report them. What were the strongest loaded variables on PC2?
Table 3: I’m unsure where the “selected variables” comes from? What were they selected for? Unselected variables like Aspect appear in Figure 3 so it can’t be that. Also, “correction” should be “correlation”. Finally, why is ggplot, an R package for making figures, being cited in a table?
255-256: Yes, Shannon’s index does show lower values than simply counting species. That’s not remarkable, that’s part of the calculation. If you think of it in terms of Hill numbers, even the exponent of Shannon’s (Hill order 1) is less than species count (Hill order 0).
267-268: This seems completely at odds with the nMDS in Fig 4. If anything, I would have thought the two groupings would have been S1, S3 and S4 versus S2. So either the stress in the nMDS plot is very high (so the point positions are unreliable) or the authors have mislabeled the plot. In either case, this needs to be re-examined.
304-307: This is an awkward sentence that doesn’t make sense. “fit quite well with” and “known for this lack” are poor language choices, and the sentence stats with “however” but doesn’t make a conflicting statement to the previous sentence.
338-340: The authors can test this assumption directly, as they have tree cores from each site that are mentioned in the Methods but not in results or discussion.
343-344: This is a incorrect conclusion to draw from the nMDS plot. Stand 3 is a subset of the Stand 4 convex hull, and the three stand 3 points are much closer to two Stand 4 points than several other points in Stand 4.
348-349: This statement seems to be referencing a known feature of the study species, but there are no citations for it.
Figure 3: Is there a reason why there is no stand information for each site? Why are there numbers 1-18 for each point? This information is meaningless. What I’d much rather see is what stand each point came from, Ideally using the symbology from Fig 4. For example, do points 1, 2 and 3 come from the same stand? That would be meaningful and tell me a lot of it as an outlier stand. But I can’t tell that from the figure.
Figure 4: It’s common in nMDS plots to indicate the stress, either in the plot or caption. The stress shows the proportion of between-site dissimilarity that was lost by coercing the points into two dimensions, and tells readers how reliable the nMDS plot is.
Reviewer 2 Report
Dear Authors,
I have reviewed the paper and found it an interesting paper.
General comments
This research deals with important aspects relative to the tree structure, biodiversity and environmental preferences of stands located on Magellanic islands within Kawésqar National Park, Chile.
The findings show the below:
An environmental matrix using databases WorldClim, SoilGrid and local topography variables were used to identify the main biophysical variables that may explain its presence. PCA was used to reduce the biophysical variables to main dimensions. PERMANOVA and nMDS were used to evaluate differences among communities. Kruskal-Wallis’ analysis was used to test differences among stand structural attributes and diversity. First axis was related to soil variables, second axis was related to climate variables and third axis to topographic variables. PERMANOVA showed separation into two main groups, which were effectively verified by nMDS’ biplot. These forest communities share the fact that they can persist over time due to the high-water table that limits the competitive effect from other tree species less tolerant to high soil water table and organic matter. These results contribute to knowledge of the species’ environmental preferences and to design conservation programs.
Comments
- Moderate English changes required.
- Cross-reference all of the citations in the text with the references in the reference section.
- Make sure that all references have a corresponding citation within the text and vice versa.
- Double-check the spelling of the author names, dates, and make sure they are correct and consistent with the citations.
- Spell out all journal titles in the reference section.
Round 2
Reviewer 1 Report
The manuscript has been much improved, and I only have minor spelling/grammar suggestions:
Line 59: "of" should be "to"
86: "transitioning" should be "transition"
137: "displayed" should be "sampled"
153: "Syx, and residual structure" should be "standard deviation and structure of residuals"
166: "of" should be "in"
178: "was" should be "were"
210: "correlations" should be "correlation"
222: "Adonis" should be "adonis", "Vegan" should be "vegan"
285 & 290: "relate" should be "related"
Fig 3: Why are the points (circles, triangles etc) different sizes on this plot?
308-309: remove the "the"s added in blue
329: remove "the" in "the Stand 2"
404: "the climate factor" should be "climate factors"
419: "is" should be "was"
429 & 430: remove "being" in "being consistent"
462: replace "it" with "conditions"
463: remove "conditions" and "can be found"
